# Epidemiological and clinical characteristics of Peruvian patients with mpox: A systematic review and meta-analysis

Darwin A. León-Figueroa[1,2], Edwin Aguirre-Milachay[3], Milagros Diaz-Torres[1], Virgilio E. Failoc-Rojas[4], Rodrigo Camacho-Neciosup[5], Abel Eduardo Chávarry Isla[6], Mario J. Valladares-Garrido[7] *

1 Facultad de Medicina Humana, Universidad de San Martín de Porres, Chiclayo, Peru, 2 Hospital Nacional Sergio E. Bernales, Lima, Peru, 3 Hospital Nacional Almanzor Aguinaga Asenjo, EsSalud, Chiclayo, Peru, 4 Universidad San Ignacio de Loyola, Lima, Peru, 5 Sociedad Científica de Estudiantes de Medicina, Universidad Nacional Pedro Ruiz Gallo, Lambayeque, Peru, 6 Escuela de Medicina, Universidad Cesar Vallejo, Chiclayo, Peru, 7 Escuela de Medicina Humana, Universidad Señor de Sipán, Chiclayo, Peru

* vgarrido@uss.edu.pe

## Abstract

### Background

Mpox has become a public health problem due to its rapid evolution and clinical variability. In Latin America, Peru ranks fifth in terms of the number of cases. The main objective of this study is to determine the epidemiological and clinical characteristics of Peruvian patients diagnosed with mpox, providing a detailed view of the situation of this affected population.

### Methods

A systematic review and meta-analysis of studies on mpox in Peru was carried out using ten databases and search tools (PubMed, Scopus, Web of Science, Embase, ScienceDirect, Google Scholar, Virtual Health Library, Scielo, Dimensions, and Epistemonikos) until August 22, 2024. The MeSH (Medical Subject Headings) terms used in the search were "mpox" and "Peru", combined with the logical operators AND and OR. Study quality was assessed using the Joanna Briggs Institute (JBI) assessment tool, and pooled estimates were generated using random-effects meta-analysis. Heterogeneity was assessed using the I² statistic. Statistical analysis was performed in R version 4.2.3, and the study was registered in PROSPERO (CRD42024582854).

### Results

A total of 150 articles were evaluated, of which 9 studies were included: four retrospective observational studies, four case series, and one case report, covering a

**Data availability statement:** "All relevant data are within the paper and its Supporting Information files."

**Funding:** The author(s) received no specific funding for this work.

**Competing interests:** The authors have declared that no competing interests exist.

total of 3960 Peruvian patients with mpox. The quality of the studies was moderate. The combined results show that 97% (95% CI: 96–98%; 3804 participants; 4 studies; $I^2 = 24\%$, p = 0.27) of the patients were male, 63% (95% CI: 57–68%; 2366 participants; 4 studies; $I^2 = 66\%$, p = 0.03) had HIV (human immunodeficiency virus), and 91% (95% CI: 83–97%; 2019 participants; 4 studies; $I^2 = 90\%$, p < 0.01) of these were receiving antiretroviral therapy. In addition, 61% (95% CI: 46–75%; 2295 participants; 4 studies; $I^2 = 95\%$, p < 0.01) identified as homosexual, and 17% (95% CI: 09–26%; 702 participants; 4 studies; $I^2 = 91\%$, p < 0.01) had a history of syphilis. The most common clinical manifestations were skin lesions (88%; 95% CI: 79–94%; 3114 participants; 4 studies; $I^2 = 92\%$, p < 0.01), lymphadenopathy (83%; 95% CI: 25–100%; 3623 participants; 2 studies; $I^2 = 100\%$, p < 0.01), anogenital rash (72%; 95% CI: 65–79%; 2657 participants; 3 studies; $I^2 = 74\%$, p = 0.02), fever (67%; 95% CI: 59–76%; 2587 participants; 4 studies; $I^2 = 86\%$, p < 0.01), and headache (52%; 95% CI: 47–57%; 1962 participants; 3 studies; $I^2 = 60\%$, p = 0.08).

## Conclusion

This systematic review provides a detailed overview on the epidemiology and clinical characteristics of Peruvian patients with mpox, highlighting a high prevalence in men and a remarkable association with HIV coinfection. The data highlight the vulnerability of the affected population and the importance of a multidisciplinary medical approach, with emphasis on early diagnosis of the most common symptoms. The findings support the implementation of prevention strategies tailored to the most vulnerable populations, especially those with HIV coinfection, and the conduct of longitudinal studies to better understand the disease.

## 1. Introduction

Mpox has emerged in recent years as a disease of global importance, raising concern among public health authorities because of its potential for spread and the clinical features it presents [1]. Since its initial diagnosis in 1970 in the Democratic Republic of Congo (DRC), mpox has spread to other regions of Africa (mainly West and Central Africa), and in recent years, it has spread to both endemic and non-endemic countries around the world [2].

On August 14, 2024, the World Health Organization (WHO) declared mpox an international public health emergency due to the rapid spread of a new strain of the Ib clade virus in the Democratic Republic of Congo, its spread to neighboring countries, and the risk of it spreading beyond Africa [3].

According to the 48th Situation Report on the multinational mpox outbreak published by WHO, 129,172 cases have been confirmed in 130 countries between January 1, 2022, and January 31, 2025 [4]. In Latin America, Peru ranks fifth, with 3,948 confirmed cases reported by the Pan American Health Organization, covering the period from January 1, 2022, to March 12, 2025 [5].

Mpox is a zoonotic orthopoxvirus, caused by a double-stranded DNA virus, which manifests mainly through skin rashes [6,7]. Two genetic clades have been identified: clade I (Congo Basin; Ib) and clade II (West Africa; IIa and IIb) [8,9]. Clade I of the virus is associated with more severe clinical symptoms and a higher case fatality rate (10.6%) compared to clade II, representing a higher public health risk [9].

Mpox has a complex and evolving nature, characterized by varied manifestations affecting multiple body systems (cutaneous, cardiovascular, oral, ophthalmic, gastrointestinal, respiratory, and pregnancy-related) [7]. However, its prodromal phase includes fever, lymphadenopathy, fatigue, and malaise, with an incubation period ranging from 5 to 21 days [10,11]. Most cases of mpox have been detected in homosexuals, bisexuals, and men who have sex with men (MSM) [12].

Transmission of mpox involves multiple routes, with sexual or intimate contact being the main mode of transmission in recent outbreaks [13,14]. However, it is also important to consider other forms of transmission, such as exposure to respiratory particles, piercings, tattoos, contaminated surfaces and objects, as well as fomites [15]. In addition, it has been shown that reinfection by mpox is possible, even in a relatively short period of time [16]. Currently, wider vaccination coverage is being promoted; 61% of the population intends to be vaccinated against mpox [17], and the effectiveness of a single dose of the JYNNEOS vaccine is 78.23% [18].

The current literature on mpox in various regions is extensive; however, there is a notable paucity of studies specifically detailing the clinical and epidemiological manifestations of this disease in Peru. This study performs a systematic review and meta-analysis to analyze the epidemiological and clinical characteristics of Peruvian patients with mpox. Through this analysis, we identify relevant patterns and differences in the manifestation of the disease compared to other regions, which will facilitate a better understanding of the current situation and allow us to provide evidence-based recommendations to optimize the clinical management and care of mpox in the country.

## 2. Materials and methods

### 2.1. Protocol and registration

The current study was conducted in accordance with the Preferred Reporting Items for Systematic Reviews and Meta-Analyses (PRISMA) guidelines (S1 Table) [19]. Additionally, the research protocol was registered with the Prospective International Registry of Systematic Reviews (PROSPERO) under the identification number CRD42024582854. The protocol originally registered in PROSPERO underwent some minor changes. These changes included specifying a detailed descriptive extraction for the reports and case series and assigning a greater number of investigators to various study processes, such as evaluation, article selection, and data analysis, all with the aim of strengthening the research.

### 2.2. Eligibility criteria

Studies that met the following criteria were included: a) observational studies (cohort, case-control, cross-sectional), case reports, and case series; b) investigations that provide information on the epidemiological and clinical characteristics of Peruvian patients with mpox; and c) patients diagnosed with mpox through clinical criteria or by RT-PCR (reverse transcriptase polymerase chain reaction). Studies that met the following criteria were excluded: editorials, letters to the editor, randomized clinical trials, conference abstracts, and narrative or systematic reviews.

### 2.3. Information sources and search strategy

Comprehensive searches were conducted in ten databases or search tools: PubMed, Scopus, Web of Sciences, Embase, ScienceDirect, Google Scholar, Virtual Health Library, Scielo, Dimensions, and Epistemonikos, until August 22, 2024, without applying language or time restrictions. The MeSH (Medical Subject Headings) terms used in the search were "mpox" and "Peru," combined using the logical operators AND and OR. The search strategy, independently validated by two authors (RCN and AECI), is detailed in S2 Table. In addition, complementary methods were used, such as manual

searches in national journals and review of the reference lists of the selected studies. However, the potential studies identified were found to be within the scope of the main strategy applied.

## 2.4. Study selection

The search results were stored using EndNote version X9 software (Thomas Reuters, New York, NY, USA). Duplicate entries, along with repeated titles and abstracts, were subsequently removed. Following this, the remaining titles and abstracts were independently screened to evaluate their alignment with the inclusion criteria. Full-text articles were then meticulously reviewed to confirm compliance with these criteria. Any discrepancies that arose during the process were resolved through consensus.

## 2.5. Outcomes

The primary outcome is to assess the prevalence of epidemiological and clinical characteristics among Peruvian patients diagnosed with mpox.

## 2.6. Quality assessment

The quality and potential bias of the studies included in the meta-analysis were evaluated using the JBI-MAStARI (Joanna Briggs Institute Meta-Analysis of Statistics Assessment and Review Instrument) tool. The assessment considered several factors, such as the research setting, outcome measures, explanatory variables, defined inclusion criteria, measurement methods, clarity in problem definition, and the rigor of statistical analysis. Based on their scores, the studies were classified as high quality (≥ 7 points), moderate quality (4–6 points), or low quality (< 4 points) (S3 Table) [20].

## 2.7. Data collection process and data items

The article data was organized in an Excel spreadsheet during the months of September and October 2024. Two authors (DALF and MDT) independently and manually extracted a comprehensive dataset. For the meta-analysis, the following variables were collected: author, year, study type, region, sample size, sex (male/female), age, mpox diagnostic method, HIV (human immunodeficiency virus) status, sexual orientation (heterosexual, homosexual, bisexual), history of syphilis, hospitalization, HIV-infected individuals receiving antiretroviral therapy, sexual risk behaviors, data collection methods, fever, headache, fatigue or asthenia, local lymphadenopathy, generalized lymphadenopathy, any type of lymphadenopathy, rash or skin lesions (local and general), anogenital rash, proctitis, lesion location, lesion morphology, and other symptoms.

For the descriptive analysis of case series and reports, the following data were extracted: author, year, study type, region, sample size, sex (male/female), age (in years), disease duration, HIV status, history of syphilis, antiretroviral therapy, sexual orientation, clinical manifestations, lesion types, lesion distribution, diagnostic method, and disease progression.

During a meeting, the data extractions conducted by the two independent authors (EAM and MJVG) were compared, and any discrepancies were resolved through mutual agreement. Following this, a thorough review and verification process was performed by a third independent investigator (VEFR) to ensure the accuracy and quality of the extracted data.

## 2.8. Data analysis

A prevalence meta-analysis (proportions) was conducted using R software version 4.2.3 (https://www.r-project.org/) (S5 Table and S6 Table). To estimate the combined prevalence of epidemiological and clinical characteristics in Peruvian patients with mpox, a variance-weighted inverse random effects model was applied. Between-study variability was assessed using the Cochrane Q statistic, while heterogeneity was quantified with the Inconsistency Index ($I^2$). Heterogeneity levels were categorized as low (<25%), moderate (25% to 50%), and high (>75%) [21].

To assess the potential presence of publication bias, two methods were employed: visual examination of the funnel plot and Egger's test. These approaches were utilized only when the meta-analysis included a minimum of 10 studies, as fewer studies reduce the test's ability to detect true asymmetry. Publication bias was considered significant if the p-value was below 0.05 [22].

The findings of the study were presented through tables and descriptive graphs. A forest plot was used to visually represent the combined prevalence of epidemiological and clinical characteristics among Peruvian patients with mpox, incorporating 95% confidence intervals to ensure a more precise depiction of the data.

## 3. Results

### 3.1. Study selection

The search strategy initially produced 150 results. After removing duplicates, 47 articles were screened by comparing their titles and abstracts against the inclusion criteria. Following this, 25 full-text articles were thoroughly evaluated, leading to the final inclusion of 9 studies in the review (S4 Table). The selection process is visually summarized in Fig 1 using a PRISMA flow chart [23–31].

### 3.2. Characteristics of the included studie

The review included four retrospective observational studies, four case series, and one case report, encompassing a total of 3,960 Peruvian patients diagnosed with mpox, aged between 20 and 50 years [23–31]. Mpox assessments were conducted using clinical evaluation and PCR testing. The studies were primarily based in the regions of Lima and La Libertad (Table 1 and 2) [23–31].

In the five case series studies and reports developed in Lima during 2022–2024, 22 Peruvian patients diagnosed with mpox were included. Of these, 90.9% (n = 20) were men and 9.1% (n = 2) were women, with a mean age of 35.8 years [27–30]. Eleven patients with HIV infection were identified, of whom two were receiving antiretroviral therapy and two had a history of syphilis. In terms of sexual orientation, seven patients identified as homosexual, five as bisexual, and seven as heterosexual. 95.5% (n = 21) of mpox cases were confirmed by PCR testing. Most patients experienced complete recovery, with a total of five deaths reported. The most common clinical manifestations included fever, lymphadenopathy, headache, and myalgia, while lesions observed consisted of vesicles, papules, ulcers, and scabs (Table 3) [27–31].

### 3.3. Quality of the included studies and publication bias

The quality of the studies included in the analysis was rated as moderate, as indicated in S3 Table [23–31]. Due to the meta-analysis comprising fewer than 10 studies, publication bias could not be evaluated using visual inspection of the funnel plot or Egger's test.

### 3.4. Prevalence of epidemiological and clinical characteristics of Peruvian patients with mpox

The overall sex distribution showed a prevalence of 97% (95% CI: 96–98%) in males and 3% (95% CI: 2–4%) in females [23–26]. Regarding medical history, syphilis was reported in 17% (95% CI: 9–26%) of cases, HIV in 63% (95% CI: 57–68%), and 91% (95% CI: 83–97%) of individuals with HIV were receiving antiretroviral therapy [23–26]. Additionally, 5% (95% CI: 3–9%) of patients required hospitalization [23–26]. Based on sexual orientation, 21% (95% CI: 11–33%) identified as heterosexual, 61% (95% CI: 46–75%) as homosexual, and 16% (95% CI: 14–17%) as bisexual [23–26]. The most common clinical manifestations included a rash or skin lesions at consultation (general), present in 88% (95% CI: 79–94%) of patients [23–26], lymphadenopathy any type in 83% (95% CI: 25–100%) [23,24], anogenital rash in 72% (95% CI: 65–79%) [23–25], fever in 67% (95% CI: 59–76%) [23–26], and headache in 52% (95% CI: 47–57%) [23,24,26] (Table 4 and Fig 2).

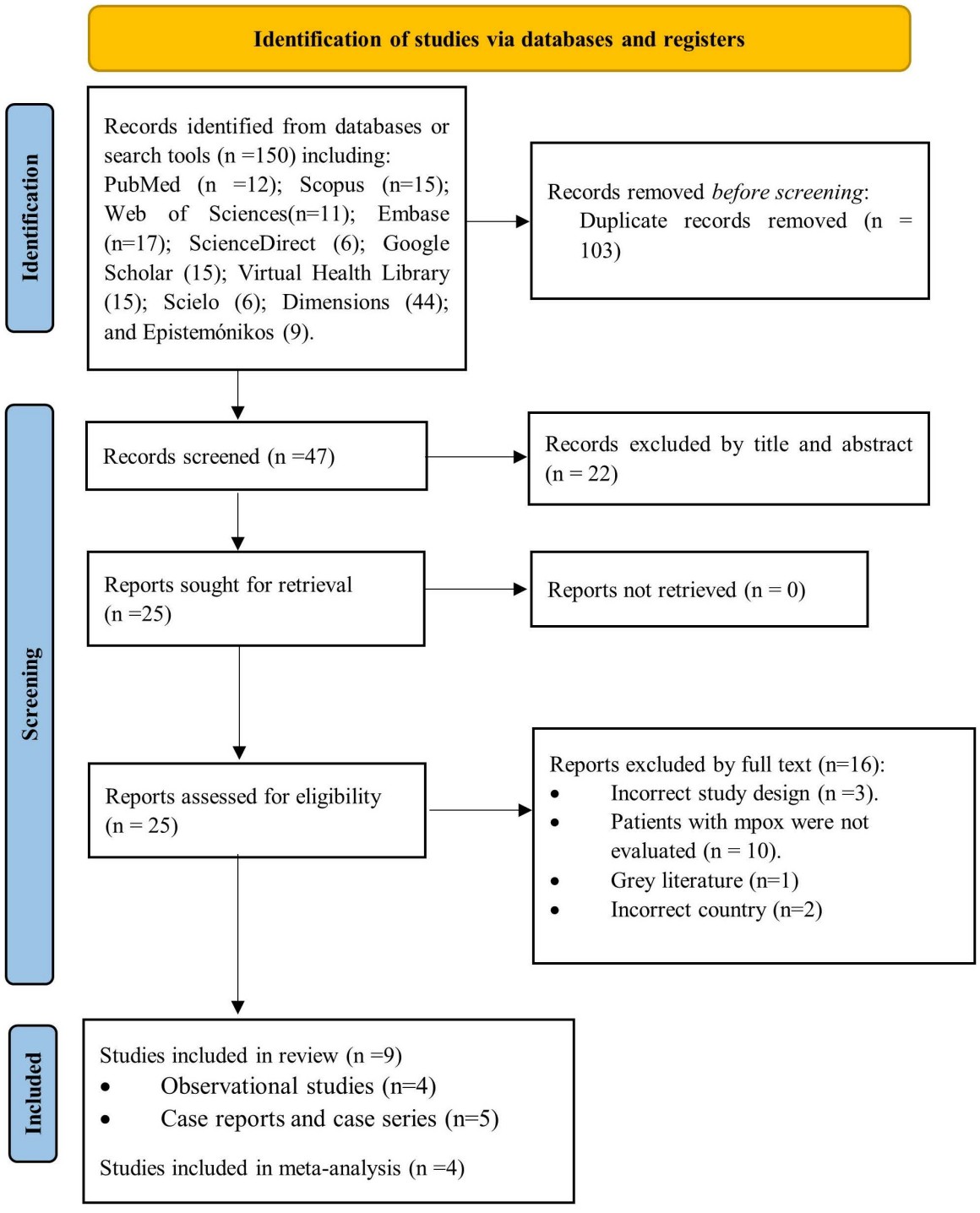

**Fig 1. Study selection process based on the PRISMA flowchart.**

**Table 1. Summary of the epidemiological characteristics of the studies included in the meta-analysis.**

| Authors | Year | Studio Type | Region | Sample | M/F | Age (years) | Mpox diagnostic method | Individuals with HIV infection | Sexual orientation | | | Syphilis antecedent | Hospitalization | HIV-infected individuals receiving antiretroviral therapy | Sexual risk behavior | Data collection methods |
|---|---|---|---|---|---|---|---|---|---|---|---|---|---|---|---|---|
| | | | | | | | | | Heterosexual | Homosexual | Bisexual | | | | | |
| Ramírez-Soto MC, et al. [23] | 2024 | Retrospective observational | Peru | 3561 | 3433/128 | Median: 32 (27–38) | Clinical and epidemiological criteria | 2123 (60%) | 821 (23%) | 2046 (57.5%) | 570 (16%) | 619 (17.4%) | 192 (5.4%) | 1796/2123 (84.6%) | Sex workers (63; 1.8%) | CDC-Peru |
| Sihuincha Maldonado M, et al. [24] | 2023 | Retrospective observational | Lima | 205 | 202/3 | Median: 32 (28–38) | Clinical and epidemiological criteria | 136 (66.3%) | 13 (6.3%) | 166 (81%) | 26 (12.7%) | 67 (32.7%) | 21 (10.2%) | 129/136 (94.8%) | Sexual encounter in the last 21 days (179; 87.3%) | Medical records |
| Alfaro Angulo MA, et al. [25] | 2024 | Retrospective observational | La Libertad | 48 | 47/1 | Range: 20–59 | PCR | 36 (75%) | 13 (27.08%) | 28 (58.33%) | 7 (14.59%) | 4 (8.33%) | 3 (6.25%) | 36/36 (100%) | Sex Club (2; 4.17%) | Medical records |
| Reaño Tovar FM, et al. [26] | 2024 | Retrospective observational | Lima | 124 | 122/2 | Range: 19–54 | PCR | 71 (57.26%) | 40 (32.26%) | 55 (44.35%) | 21 (16.94%) | 12 (9.68%) | 2 (1.61%) | 58/71 (81.69%) | Men who have sex with other men (76; 61.29%) | Medical records |

NS: Not specified; NR: Not Reported; M/F: Male/ Female; PCR: Polymerase chain reaction; CDC-Peru: National Center for Epidemiology, Disease Prevention, and Control of Peru.

**Table 2. Summary of the clinical characteristics of the studies included in the meta-analysis.**

| Authors | Year | Fever | Headache | Fatigue or Asthenia | Local lymphade-nopathy | General lymphade-nopathy | Lymphade-nopathy any type | Rash or skin lesions at consulta-tion (Local) | Rash or skin lesions at con-sultation (General) | Anogeni-tal rash | Proctitis | Lesion location | Lesion morphology | Other symptoms |
|---|---|---|---|---|---|---|---|---|---|---|---|---|---|---|
| Ramírez-Soto MC, et al. [23] | 2024 | 2317 (65%) | 1780 (49.9%) | 1095 (30.7%) | 1307 (36.7%) | 209 (5.9%) | 3512 (98.6%) | 750 (21%) | 2792 (78.4%) | 2464 (69.2%) | 2464 (69.2%) | NS | NS | Myalgia (1367; 38.4%) |
| Sihuincha Maldo-nado M, et al. [24] | 2023 | 162 (79%) | 119 (58.1%) | 105 (51.5%) | 98 (88.3%) | 13 (11.7%) | 111 (54.2%) | 38 (18.6%) | 166 (81.4%) | 160 (78.0%) | 19 (9.3%) | Face, head or neck (128; 62.7%), Trunk (142; 69.3%), Upper extremities (111; 54.4%), and Lower extremities (86; 41.9%) | Macular (116; 56.6%), Papular (162; 79%), Vesicle (44; 21.5%), Pustule (177; 86.3%), Ulcer (4; 2%), and Crust (65; 31.7%) | Cough (7; 3.4%), Rhinorrhea (6; 2.9%), Sore throat (79; 38.5%), Malaise (123; 60%), Back pain (77; 37.6%), Chills and sweats (69; 33.8%), and Pruritus (27; 13.2%) |
| Alfaro Angulo MA, et al. [25] | 2024 | 26 (54.17%) | NR | 25 (52.08%) | NS | 25 (52.08%) | Inguinal (12; 25%), cervical (8; 16.67%), and axillary (5; 10.42%). | NS | 48 (100%) | 33 (68.75%) | 3 (6.25%) | Face (40; 83.33%), Chest (37; 77.08%), Mouth or lips (12; 25%), and Extrem-ities (11; 22.92%). | NR | Myalgia (21; 43.75%), back pain (21; 43.75%), sore throat (18; 37.5%), and chills (5; 10.42%). |
| Reaño Tovar FM, et al. [26] | 2024 | 82 (66.13%) | 63 (50.81%) | 25 (20.16%) | 19 (15.32%) | 9 (7.26%) | NS | 15 (12.10%) | 108 (87.10%) | NS | 6 (4.84%) | NS | Crust (103; 83.06%), Pustule (66; 53.22%), Papule (15; 12.09%), Vesi-cle (3; 2.41%), and Macule (2; 1.61%). | Chills (57; 45.97%), myalgia (35; 28.23%), backache (25; 20.16%), diar-rhea (2; 1.61%), and pruritus (1; 0.81%). |

NS: Not specified; NR: Not Reported.

**Table 3. Summary of the epidemiological and clinical characteristics of the reports and case series included in the systematic review.**

| Authors | Year | Studio Type | Region | Sample | M/F | Age (years) | Time of illness | Individuals with HIV infection | Syphilis antecedent | Antiretroviral therapy | Sexual orientation | Clinical manifestations | Types of lesions | Distribution | Diagnostic method | Evolution |
|---|---|---|---|---|---|---|---|---|---|---|---|---|---|---|---|---|
| Pampa-Espinoza L, et al. [27] | 2022 | Case series | Lima | 2 | M | 58 | 9 days | Yes | Non-reactive | Yes | NS | Fever, inguinal-cervical lymphadenopathy, and odynophagia. | Vesicles, pustules, ulcers, and crusts | Generalized centrifuge | PCR | Isolation and monitoring at CDC Peru |
| | | | | | M | 33 | 8 days | Yes | Non-reactive | Yes | NS | Fever, right inguinal adenopathy | Vesicles and pustules | Non-generalised centrifuge | PCR | Isolation and monitoring at CDC Peru |
| Bonifacio Morales N, et al. [31] | 2024 | Case series | Lima | 3 | M | 32 | NR | No | No | No | Heterosexual | Fever, headache, ocular pain, ocular pruritus, photophobia, blurred vision, lacrimation, and edematous ocular congestion. | Vesicles and pustules | Generalized | Clinical and epidemiological criteria | Recovered |
| | | | | | F | 27 | NR | No | No | No | Heterosexual | Myalgia, fever, odynophagia, and anal pain with bleeding during bowel movements. | Vesicles and pustules | Generalized | PCR | Recovered |
| | | | | | M | 8 | NR | No | No | No | Heterosexual | Odynophagia, tinnitus, fever, general malaise, and cervical adenopathy. | Vesicles and pustules | Generalized | PCR | Recovered |
| Terry Castellano LE, et al. [28] | 2023 | Case report | Lima | 1 | M | 39 | 5 days | Yes | NR | No | Homosexual | Pustular lesions, anorectal pain, fecal incontinence, and Fournier's perianal fasciitis. | Pustules | Generalized | PCR | Died |
| Briceño M. [29] | 2023 | Case series | Lima | 12 | M | 37 | 3 days | Yes | NR | NR | Homosexual | Fever, headache, malaise | Vesicle | Anus and hands | PCR | Recovered |
| | | | | | M | 45 | 4 days | Yes | NR | NR | Homosexual | Fever, headaches, and lymphadenopathies | Macule, papule, vesicle, and pustule | Face, thorax, extremities, and genitalia | PCR | Recovered |
| | | | | | M | 38 | 2 days | NR | NR | NR | Heterosexual | Fever, headache, odynophagia, and back pain | Papule, vesicle, pustule, and scab | Face, mouth, extremities, and genitalia | PCR | Recovered |

(Continued)

**Table 3.** (Continued)

| Authors | Year | Studio Type | Region | Sample | M/F | Age (years) | Time of illness | Individuals with HIV infection | Syphilis antecedent | Antiretroviral therapy | Sexual orientation | Clinical manifestations | Types of lesions | Distribution | Diagnostic method | Evolution |
|---|---|---|---|---|---|---|---|---|---|---|---|---|---|---|---|---|
| | | | | | M | 36 | 1 day | NR | NR | NR | Bisexual | Fever, chills, headache, asthenia, and myalgia | Papule, vesicle, and pustule | Chest, extremity, and genitalia | PCR | Recovered |
| | | | | | M | 32 | 1 day | NR | NR | NR | Bisexual | Fever, chills, myalgia, and odynophagia | Macule, papule, vesicle, and scab | Face, thorax, extremity, and genitalia | PCR | Recovered |
| | | | | | M | 39 | NR | Yes | NR | NR | Homosexual | NR | Papule, vesicle, and crust | Face, thorax, and extremity | PCR | Recovered |
| | | | | | M | 26 | 3 days | NR | Yes | NR | Bisexual | Fever, chills, headache, odynophagia, and myalgia | Papule | Face, mouth, thorax, and extremities | PCR | Recovered |
| | | | | | M | 29 | 2 days | Yes | NR | NR | Bisexual | Fever, headache, and myalgia | Vesicle and pustule | Face, thorax, extremity, and genitalia | PCR | Recovered |
| | | | | | M | 39 | 1 day | NR | NR | NR | Heterosexual | Fever, chills, headache, asthenia, myalgia, and lymphadenopathy | Vesicle, pustule, and crust | Face, thorax, and extremities | PCR | Recovered |
| | | | | | M | 34 | 3 days | NR | NR | NR | Heterosexual | Chills, headache, myalgia, and backache | Vesicle and pustule | Face, mouth, thorax, extremities, and genitalia | PCR | Recovered |
| | | | | | F | 48 | 1 day | NR | NR | NR | Heterosexual | Fever, chills, odynophagia, and lymphadenopathy | Papule, vesicle, pustule, and scab | Face, thorax, extremity, and genitalia | PCR | Recovered |
| | | | | | M | 29 | 2 days | NR | NR | NR | Bisexual | Fever, headache, odynophagia, and asthenia | Macule, pustule, and crust | Face, thorax, and genitalia | PCR | Recovered |

*(Continued)*

| Authors | Year | Studio Type | Region | Sample | M/F | Age (years) | Time of illness | Individuals with HIV infection | Syphilis antecedent | Antiretroviral therapy | Sexual orientation | Clinical manifestations | Types of lesions | Distribution | Diagnostic method | Evolution |
|---|---|---|---|---|---|---|---|---|---|---|---|---|---|---|---|---|
| **Araujo-Castillo JF, et al. [30]** | 2023 | Case series | Lima | 4 | M | 41 | 18 days • | Yes | No | No | Homosexual | Fever, headache, asthenia, sore throat, lymphadenopathy, and proctitis. | Polymorphic | Generalized | PCR | Died |
| | | | | | M | 32 | 32 days • | Yes | Yes | No | Homosexual | Fever, asthenia, sore throat, proctitis, and dysphagia. | Polymorphic | Generalized | PCR | Died |
| | | | | | M | 23 | 18 days • | Yes | No | No | Homosexual | Fever, chills, headache, back pain, and lymphadenopathy. | Monomorphic | Generalized | PCR | Died |
| | | | | | M | 23 | 47 days • | Yes | No | No | NR | Fever, headache, and lymphadenopathy. | Polymorphic | Localized | PCR | Died |

NS: Not specified; NR: Not Reported; M/F: Male/ Female; PCR: Polymerase chain reaction; CDC-Peru: National Center for Epidemiology, Disease Prevention, and Control of Peru. • Number of days elapsed in the clinical evolution of those who died of mpox.

**Table 4. Pooled prevalence of epidemiological and clinical characteristics of Peruvian patients with mpox.**

| | Studies | Cases | Sample size | I2 (%) | p-value | Prevalence % (95% CI) | Supporting information |
|---|---|---|---|---|---|---|---|
| **Sex** | | | | | | | |
| Male | 4 | 3804 | 3938 | 24% | p=0.27 | 97 (96 - 98) | S1 Fig |
| Female | 4 | 134 | 3938 | 24% | p=0.27 | 3 (02 - 04) | S2 Fig |
| Antecedents | | | | | | | |
| Syphilis | 4 | 702 | 3938 | 91% | p<0.01 | 17 (09 - 26) | S3 Fig |
| Individuals with HIV | 4 | 2366 | 3938 | 66% | p=0.03 | 63 (57 - 68) | S4 Fig |
| Individuals with HIV on HAART | 4 | 2019 | 2366 | 90% | p<0.01 | 91 (83 - 97) | S5 Fig |
| Hospitalization | 4 | 218 | 3938 | 75% | p<0.01 | 05 (03 - 09) | S6 Fig |
| Sexual orientation | | | | | | | |
| Heterosexual | 4 | 887 | 3938 | 94% | p<0.01 | 21 (11 - 33) | S7 Fig |
| Homosexual | 4 | 2295 | 3938 | 95% | p<0.01 | 61 (46 - 75) | S8 Fig |
| Bisexual | 4 | 624 | 3938 | 0% | p<0.01 | 16 (14 - 17) | S9 Fig |
| Clinical manifestations | | | | | | | |
| Fever | 4 | 2587 | 3938 | 86% | p<0.01 | 67 (59 - 76) | S10 Fig |
| Headache | 3 | 1962 | 3890 | 60% | p=0.08 | 52 (47 - 57) | S11 Fig |
| Myalgia | 3 | 1423 | 3733 | 67% | p=0.05 | 36 (29 - 44) | S12 Fig |
| Fatigue or Asthenia | 4 | 1250 | 3938 | 94% | p<0.01 | 37 (25 - 51) | S13 Fig |
| Local lymphadenopathy | 3 | 1424 | 3890 | 95% | p<0.01 | 33 (20 - 47) | S14 Fig |
| General lymphadenopathy | 4 | 256 | 3938 | 95% | p<0.01 | 13 (05 - 25) | S15 Fig |
| Lymphadenopathy any type | 2 | 3623 | 3766 | 100% | p<0.01 | 83 (25 - 100) | S16 Fig |
| Rash or skin lesions at consultation (Local) | 3 | 803 | 3890 | 72% | p=0.03 | 18 (13 - 23) | S17 Fig |
| Rash or skin lesions at consultation (General) | 4 | 3114 | 3938 | 92% | p<0.01 | 88 (79 - 94) | S18 Fig |
| Anogenital rash | 3 | 2657 | 3814 | 74% | p=0.02 | 72 (65 - 79) | S19 Fig |
| Proctitis | 4 | 2492 | 3938 | 100% | p<0.01 | 19 (00 - 65) | S20 Fig |

Note: Confidence interval (CI).

## 4. Discussion

### Epidemiological trends

The present study provides a comprehensive overview of the epidemiological and clinical characteristics of Peruvian patients with mpox, providing relevant evidence in the Latin American context. In terms of epidemiology, we found that males accounted for 97% of cases, a finding consistent with recent global literature on mpox outbreaks that reports a marked male predominance, particularly among MSM [32,33]. This pattern has been consistent in outbreaks in Colombia [34] and regions such as Europe [35] and North America [36], suggesting that factors such as sexual contact networks and risk practices may be driving transmission in these specific populations. On the other hand, transgender women may have a higher risk of sexual transmission compared to cisgender women, coexisting with HIV infection and lesions similar to those found in men, such as lesions in the anogenital region [37]. This increased risk in transgender women is influenced by various factors, including societal stigma, discrimination, limited access to healthcare, and higher rates of HIV infection, which together contribute to their vulnerability to mpox [37,38].

The Table 5 presents a detailed analysis of the clinical characteristics and epidemiological background of Peruvian patients diagnosed with mpox between 2022 and 2024, based on data from the National Center for Epidemiology, Disease Prevention, and Control of Peru. It is observed that the majority of cases are concentrated in young adults (18–59

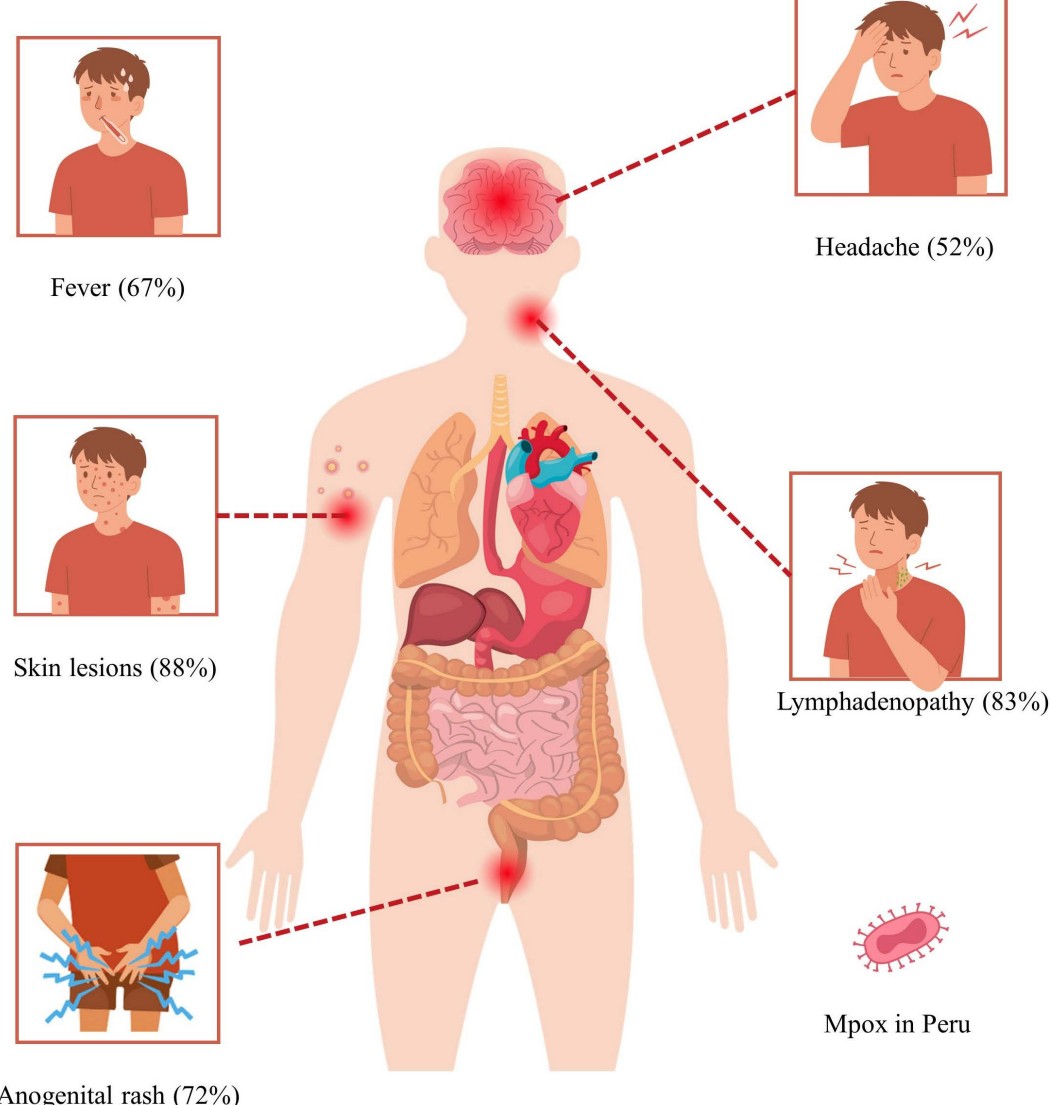

Fever (67%)

Headache (52%)

Skin lesions (88%)

Lymphadenopathy (83%)

Anogenital rash (72%)

Mpox in Peru

**Fig 2. Most prevalent clinical manifestations in Peruvian patients with mpox.**

years), with a notably higher prevalence in men (96.1%). The predominant sexual orientation among patients is homosexual (56.2%), followed by bisexual (15.5%) and heterosexual (24.3%). A significant percentage of patients live with HIV (55.6%), most of whom are undergoing antiretroviral treatment (87.9%). The most common clinical manifestations include fever, generalized rash, headache, and myalgia. In terms of background, a considerable percentage report previous infections such as syphilis (16.1%) and other infections (7.7%) [39].

These data are consistent with that reported by León-Figueroa DA et al., who identified that 98.72% of 4537 confirmed cases involved men, with 95.72% linked to MSM [6], and by WHO, which reported that 97.1% of cases during the 2022 outbreak involved young men with a median age of 35 years [40]. This epidemiological pattern is also reflected in the work of Du M. et al., who reported that 79.8% of cases involved MSM [41], underscoring that close contact and risky sexual practices are determinant factors in the transmission of the virus [13,14].

**Table 5. Characteristics and epidemiological background of Peruvian patients with mpox from 2022 to 2024, according to the Peruvian National Center for Epidemiology, Prevention, and Disease Control.**

| Variables | 2022 (n = 3698) | 2023 (n = 164) | 2024 (n = 89) | Total (3951) |
|---|---|---|---|---|
| **Clinical characteristics** | | | | |
| Age ⁑ | 32 (27- 38) | 32 (26 - 39) | 33 (28 - 37) | Median: 32 |
| Time for case identification ⁑ | 6 (4 - 8) | 6 (4 - 9) | 9 (5 - 29) | Median: 6 |
| **Sex** | | | | |
| Male | 3557 (96.2%) | 156 (95.1%) | 85 (95.5%) | 3798 (96.1%) |
| Woman | 141 (3.8%) | 8 (4.9%) | 4 (4.4%) | 153 (3.9%) |
| **Stages of life** | | | | |
| Child (0–11 years) | 3 (0.1%) | 1 (0.6%) | 0 (0%) | 4 (0.1%) |
| Adolescent (12–17 years) | 24 (0.6%) | 1 (0.6%) | 1 (1.1%) | 26 (0.7%) |
| Youth (18–29 years) | 1424 (38.5%) | 63 (38.4%) | 27 (30.3%) | 1514 (38.3%) |
| Adult (30–59 years) | 2233 (60.4%) | 99 (60.4%) | 61 (68.5%) | 2393 (60.5%) |
| Older adults (60 years and over) | 14 (0.4%) | 0 (0%) | 0 (0%) | 14 (0.4%) |
| **Sexual Orientation** | | | | |
| Homosexual | 2088 (56.5%) | 67 (40.9%) | 66 (74.1%) | 2221 (56.2%) |
| Bisexual | 563 (15.2%) | 37 (22.6%) | 13 (14.6%) | 613 (15.5%) |
| Heterosexual | 904 (24.4%) | 45 (27.4%) | 10 (11.2%) | 959 (24.3%) |
| Unknown | 143 (3.9%) | 15 (9.1%) | 0 (0%) | 158 (4%) |
| **Specific population** | | | | |
| Transgender woman | 32 (0.9%) | 3 (1.8%) | 0 (0%) | 35 (0.9%) |
| Sex worker | 57 (1.5%) | 11 (6.7%) | 3 (3.3%) | 71 (1.8%) |
| **Antecedents** | | | | |
| **People living with HIV (PLHIV)** | | | | |
| Yes | 2057 (55.6%) | 81 (49.3%) | 61 (68.5%) | 2199 (55.6%) |
| No | 1448 (39.2%) | 62 (38.8%) | 21 (23.6%) | 1531 (38.8%) |
| Unknown | 193 (5.2%) | 21 (12.8%) | 7 (7.8%) | 221 (5.6%) |
| **People living with HIV on Antiretroviral Therapy (HAART)** | | | | |
| Yes | 1815/2057 (88.2%) | 66/81 (81.5%) | 51/61 (83.6%) | 1932/2199 (87.9%) |
| No | 213/ 2057 (10.4%) | 13/81 (16.0%) | 10/61 (16.3%) | 236/2199 (10.7%) |
| Unknown | 29/2057 (1.4%) | 2/81 (2.5%) | 0/61 (0%) | 31/2199 (1.4%) |
| **History of syphilis within the last year** | | | | |
| Yes | 618 (16.7%) | 18 (10.9%) | 0 (0%) | 636 (16.1%) |
| No | 2581 (69.8%) | 108 (65.9%) | 5 (5.6%) | 2694 (68.2%) |
| Unknown | 499 (13.5%) | 38 (23.2%) | 84 (94.3%) | 621 (15.7%) |
| **Other infections** | | | | |
| Yes | 271 (7.3%) | 16 (9.8%) | 17 (19.1%) | 304 (7.7%) |
| No | 3105 (84%) | 121 (73.8%) | 47 (52.8%) | 3273 (82.8%) |
| Unknown | 322 (8.7%) | 27 (16.4%) | 25 (28.1%) | 374 (9.5%) |
| **Clinical manifestations** | | | | |
| Fever | 2394 (64.74%) | 94 (57.32%) | 3 (3.37%) | 2491 (63%) |
| Generalized rash (including anogenital region) | 2037 (55.08%) | 88 (53.66%) | 21 (23.6%) | 2146 (54.3%) |
| Headache | 1852 (50.08%) | 74 (45.12%) | 3 (3.37%) | 1929 (48.8%) |
| Myalgia | 1415 (38.26%) | 64 (39.02%) | 1 (1.12%) | 1480 (37.5%) |
| Localized lymphadenopathy | 1326 (35.86%) | 56 (34.15%) | 1 (1.12%) | 1383 (35%) |
| Asthenia | 1126 (30.45%) | 44 (26.83%) | NR | 1170 (29.6%) |
| Sore throat | 1001 (27.07%) | 38 (23.17%) | 1 (1.12%) | 1040 (26.3%) |

*(Continued)*

**Table 5.** (Continued)

| Variables | 2022 (n = 3698) | 2023 (n = 164) | 2024 (n = 89) | Total (3951) |
|---|---|---|---|---|
| Chills | 986 (26.66%) | 61 (37.2%) | NR | 1047 (26.5%) |
| Back pain | 903 (24.42%) | 29 (17.68%) | NR | 932 (23.6%) |
| Generalized rash (excludes anogenital region) | 871 (23.55%) | 28 (17.07%) | 8 (8.99%) | 907 (23%) |
| Anogenital rash only | 556 (15.04%) | 23 (14.02%) | 43 (48.31%) | 622 (15.7%) |
| Localized rash (excludes anogenital region) | 234 (6.33%) | 16 (9.76%) | 17 (19.1%) | 267 (6.8%) |
| Generalized lymphadenopathy | 212 (5.73%) | 10 (6.1%) | NR | 222 (5.6%) |
| Proctitis | 201 (5.44%) | 23 (14.02%) | NR | 224 (5.7%) |
| Other symptoms | 373 (10.09%) | 12 (7.32%) | 84 (94.38%) | 469 (11.9%) |

⁎⁎ The median [Q1; Q3] for age and time of illness is shown. NR: Not reported.

Although current data indicate a higher prevalence of mpox among men, particularly among MSM, this trend should not be interpreted as evidence of lower impact among women. A meta-analysis by Satapathy et al. revealed a notable decline in the proportion of female cases—from 44.09% in studies conducted prior to 2022 to just 2.40% in those published thereafter [42]. This proportion varies significantly depending on geographic region, endemic status of the country, and the reporting period. Such disparities point to the possibility of diagnostic biases and underreporting in women, rather than a true difference in disease burden. Therefore, strengthening epidemiological surveillance systems with an inclusive approach that considers all populations—regardless of gender—is essential. Doing so will not only provide a more accurate picture of mpox transmission dynamics but also help address the specific health needs of women and other vulnerable groups [38].

### Impact of HIV coinfection in patients with mpox

In relation to comorbidities, coinfection with HIV was observed in 63% of the cases, highlighting the high vulnerability of this population to mpox. This percentage exceeds those reported in previous studies, which documented coinfection rates between 36% and 41% [32,43,44]. According to a systematic review by Ortiz-Saavedra B. et al., 40.32% of 6345 confirmed cases of mpox were coinfected with HIV [45]. Similarly, Liu BM et al. reported that more than 35% of cases had coinfection with HIV, in addition to 40% with sexually transmitted infections (STIs), such as *Chlamydia trachomatis, Neisseria gonorrhoeae, Treponema pallidum*, and herpes simplex virus [46]. Likewise, another systematic review proposed by Chenchula S. et al. reported that 36.1% of 18,275 cases of mpox had coinfection with HIV, reinforcing the relevance of this comorbidity in the clinical course of the disease [47].

However, it is important to differentiate between individuals living with HIV who have well-controlled infections, characterized by undetectable viral loads and preserved immune function, and those with advanced HIV or AIDS, who may have significant immunosuppression, characterized by low CD4 + cell counts and high viral loads [48,49]. On the other hand, the fact that 91% of coinfected patients were receiving antiretroviral therapy suggests that, although access to treatment is adequate, these individuals remain susceptible to mpox infection, raising questions about the impact of relative immunosuppression due to higher viral loads and lower CD4 counts on the clinical course of the disease [50]. Individuals with well-controlled HIV, who have undetectable viral loads and CD4 + counts above 500 cells/mm3, may experience a less severe course of mpox compared to those with advanced HIV or AIDS, who are at higher risk for severe disease due to significant immunosuppression [50,51]. A global case series that included only HIV patients with CD4 + cell counts less than 350 cells/mm3 reported severe mpox necrotizing disease and more severe complications in HIV patients who had CD4 + cell counts less than 200 cells/mm3; on the other hand, in 21 of 85 patients who initiated or restarted ART, an inflammatory syndrome of immune reconstitution to mpox was suspected [50].

These findings could be explained by the fact that HIV causes immunosuppression, which makes it difficult for the body to mount an effective immune response against pathogens such as mpox. People who are coinfected are at increased risk of severe illness and death; it is critical to prioritize interventions and improve treatment strategies designed specifically for people living with HIV, distinguishing between those with well-controlled infections and those with advanced immunosuppression [52,53].

## Clinical manifestations of mpox patients

Our study documented a high prevalence of skin lesions (88%), lymphadenopathy (83%), anogenital rash (72%), fever (67%), and headache (52%), findings that coincide with those described in the scientific literature. Jaiswal V. et al. reported that the most significant symptoms in patients with mpox were rash (100%) and fever (99%), in addition to upper respiratory symptoms (55%), headache (78%), vomiting (25%), oral ulcers (56%), conjunctivitis (21%), and lymphadenopathy (85%) [54].

Similarly, the meta-analysis by Benites-Zapata VA. et al. noted that among 1958 patients, the most prevalent manifestations were rash (93%), fever (72%), pruritus (65%), and lymphadenopathy (62%) [55]. A meta-analysis by Satapathy P. et al. identified a broad spectrum of clinical manifestations of mpox, encompassing cutaneous, cardiovascular, oral, ophthalmic, gastrointestinal, respiratory, and pregnancy-related symptoms. Cutaneous manifestations, present in up to 100% of cases, were the most prevalent, highlighting characteristic lesions and rashes. Constitutional symptoms associated with viral diseases were reported in 60% to more than 85% of cases, while significant respiratory symptoms affected approximately 50%. Among neurological symptoms, headaches were the most common, present in more than 30% of patients. Gastrointestinal manifestations included oral lesions in 39% of cases, with less frequent diarrhea (~5%) and proctitis, especially in adolescents and young adults. Finally, ophthalmic manifestations were less common (6%), although with notable variability among the studies analyzed [7].

Other studies have reported a high prevalence of systemic symptoms in countries where mpox is endemic, indicating a prevalence between 85 and 90% for fever, 80% for headache, and between 70 and 100% for lymphadenopathy, which is higher than that reported in our country [56–58]. Regarding cutaneous and anogenital rash, they have had an important consideration in our country. An anogenital, oral, and perioral pattern has been reported, which raised the hypothesis that the lesions appear at the site of inoculation [37,58]. However, a distribution of lesions on the face, trunk, and extremity areas has also been reported [43,44,59]. These results could be explained by variations in clinical manifestations according to the severity of mpox (from mild to severe), demographic and epidemiological differences in the populations studied, the stage of the disease at the time of evaluation, as well as possible changes in the virulence of the virus or in the immune response of the patients [60,61]. It is important to consider the limitations of the cited studies, such as sample heterogeneity due to variations in patient populations, regions, and clinical settings, which can affect the generalizability of the results. Additionally, publication bias should be taken into account, as studies with positive results may be overrepresented, influencing the overall conclusions.

## Hospitalization rates and disease severity in mpox patients

Regarding disease severity and hospitalization, only 5% of patients in our review required hospitalization, a relatively low figure compared to a study in Brazil where the hospitalization rate was 10.5%, with people with HIV having more proctitis and requiring invasive support, and mpox severity was associated with low CD4 + cell counts and discontinuation of HIV treatment [62]. It is important to note that the healthcare infrastructure and diagnostic practices in Peru and Brazil may vary, with differences in the availability of resources, healthcare access, and hospitalization criteria potentially influencing the reported rates. Another study in 2022 that evaluated a cohort of patients hospitalized for mpox found that there were no clinical, laboratory, or complication differences in patients with and without HIV coinfection; however, no HIV patients had advanced disease [63]. Additionally, some complications of mpox disease have been reported, such as ocular

manifestations, coalescing lesions resulting in large plaques, encephalitis, and encephalomyelitis, among others. Also, in patients with advanced HIV infection, a necrotizing form of mpox has been reported [64,65].

A meta-analysis proposed by Benites-Zapata VA, et al., reported that among patients with mpox, 35% were hospitalized, and about 4% of hospitalized patients had fatal outcomes [55]. These results could be explained by variations in the characteristics of the populations studied, such as the prevalence of severe comorbidities (e.g., immunosuppression or advanced HIV), the level of access to health services, and the application of early diagnosis and timely management strategies in various epidemiological contexts [10,66]. The low rates observed in our population could be due to the fact that most HIV patients were receiving antiretroviral therapy, which could have mitigated the severity of infection. However, these findings should be interpreted with caution, as the small number of studies included in the analysis limits the ability to make robust comparisons.

### Strengths and limitations of the study

This study has several strengths. First, a significant sample of 3960 Peruvian patients with mpox was included, which allows us to provide a representative view of the epidemiology and clinical characteristics of the disease in the national context. This is particularly relevant at a time when information on mpox in Latin America remains limited. In addition, our analysis provides important data on HIV coinfection, a critical factor in understanding the vulnerability and clinical evolution of this specific population. Another key strength is the methodological rigor applied at all stages of the study, following PRISMA guidelines for systematic reviews and meta-analyses, which guarantees the reproducibility of the results. Likewise, the use of multiple databases, both international and regional, allowed for an exhaustive search, which reduces the risk of omitting relevant studies.

However, the study also has some limitations that should be considered when interpreting the results. First, the limited number of included studies (n = 9) restricts the generalizability of the findings and may have influenced the precision of the estimates derived from the meta-analysis. Second, the inability to assess publication bias due to the small number of studies is another important limitation, as this could have introduced undetected bias in the results. Third, most of the included studies were retrospective in nature, which may be associated with an increased possibility of bias, particularly in data collection. The retrospective nature of these studies could have led to recall bias, missing data, and unmeasured confounding variables, all of which could have influenced the reported trends. Finally, heterogeneity among the investigations, both in terms of design and patient characteristics, could have influenced the findings, making it difficult to identify consistent patterns across the populations analyzed. The wide confidence intervals in some estimates suggest a high degree of uncertainty in the aggregated results, requiring caution when interpreting them. The implications of this variability indicate that care should be taken when considering estimates with high variability. Additionally, it is recognized that more studies with larger sample sizes and more precise measurements are needed to reduce uncertainty and provide more reliable estimates.

To address some of the limitations identified in this study, future research should aim to include a larger number of prospective studies, which would reduce the risk of bias and improve the generalizability of the findings. In addition, implementing more rigorous methods to assess and control for potential confounders and conducting sensitivity analyses to evaluate the impact of missing data would enhance the reliability of the conclusions drawn. Expanding the study sample and standardizing patient characteristics across studies could help minimize heterogeneity and provide clearer insights into the epidemiological trends.

### Clinical implications

The results of this study have important clinical implications, especially in the management of patients with mpox in populations with high HIV prevalence. The high percentage of HIV coinfection among Peruvian patients addresses the need for a comprehensive approach to the diagnosis and treatment of mpox in immunocompromised individuals [46]. Recognizing

the most prevalent clinical features, such as skin lesions, fever, and lymphadenopathy, is critical to improving early diagnosis and avoiding serious complications [67]. In addition, specialized care should focus on this high-risk population, not only for the management of acute mpox symptoms but also for co-infection surveillance and long-term monitoring of their immune status. Early detection and appropriate treatment of these features can help reduce the rate of hospitalization and improve clinical outcomes [68].

Public health interventions should be a key component of the response to mpox, particularly in the context of populations with high HIV prevalence. Vaccination efforts should prioritize high-risk groups, including individuals living with HIV, as part of a comprehensive prevention strategy [17,69]. Contact tracing and active surveillance are also crucial for identifying and isolating cases early, preventing further spread. Engaging with communities, particularly through education and outreach programs, will be essential to increasing awareness of mpox, its symptoms, and the importance of timely treatment [70]. Such strategies are especially relevant in regions like Peru, where diverse populations may face varying levels of access to healthcare and public health infrastructure [70,71].

Longitudinal studies are recommended to evaluate the long-term progression of mpox in people with HIV, as well as their response to therapeutic interventions. It is also crucial to expand research to other regions of Peru, given that most of the studies included in this review focused on urban areas such as Lima and La Libertad. Research in more remote and diverse regions could provide a broader understanding of the epidemiology of mpox throughout the country. Additionally, greater inclusion of women and heterosexuals in future studies is needed, given that this review reflects an overrepresentation of homosexual and bisexual males. This will allow for a more complete and equitable view of the different population groups affected by the disease. Finally, intervention studies focused on prevention and treatment in HIV co-infected patients could also offer key strategies to mitigate the effects of future outbreaks. These studies could focus on the development of novel therapeutic approaches, as well as preventive measures tailored to the unique needs of co-infected individuals. This could include exploring the role of antiretroviral therapy in reducing the susceptibility to mpox, understanding the impact of immunocompromised states on disease progression, and evaluating the effectiveness of vaccines in this specific population. In addition, identifying biomarkers or other indicators of early infection in co-infected patients may help improve early detection and response strategies, which are essential for controlling future outbreaks.

## 5. Conclusions

This systematic review provides a comprehensive overview of the epidemiological and clinical characteristics of Peruvian patients diagnosed with mpox, highlighting the high prevalence of the disease in men and its significant correlation with HIV coinfection. The findings reveal that 63% of patients had HIV coinfection, underscoring the vulnerability of this population and the need for a multidisciplinary health care approach that addresses both mpox and associated comorbidities.

The high prevalence of clinical manifestations such as skin lesions, fever, and lymphadenopathy highlights the importance of early and accurate diagnosis, which can facilitate better care and treatment. However, the limited nature of the studies analyzed, as well as the predominance of research in urban areas, indicates that greater geographic and demographic representation is required in future research to obtain a more complete picture of the epidemiology of mpox in Peru.

The clinical implications of these findings are significant, as they emphasize the need for targeted prevention strategies aimed at high-risk populations, particularly men who have sex with men and individuals living with HIV. Public health efforts should prioritize the integration of routine HIV testing and screening for mpox in areas with high HIV prevalence to facilitate early detection and intervention. Additionally, specific educational campaigns targeting vulnerable populations are necessary to raise awareness about mpox symptoms and encourage timely medical attention. Strengthening the availability and accessibility of antiviral treatments and ensuring that healthcare professionals are adequately trained to manage co-infected individuals are also critical strategies for minimizing the impact of mpox.

Future studies, including longitudinal and interventional trials, are essential to better understand the long-term evolution of mpox in individuals with HIV and to develop effective strategies to mitigate the impact of the disease in vulnerable

populations. As we continue to face outbreaks of mpox, the integration of epidemiologic surveillance systems with public health infrastructure to monitor trends and respond rapidly to outbreaks should be prioritized. Collaboration across public health sectors and the establishment of targeted strategies based on local epidemiology will be crucial in effectively managing the disease.

## Supporting information

**S1 Table. PRISMA Checklist (PRISMA 2020 Main Checklist and PRIMSA Abstract Checklist).**
(DOCX)

**S2 Table. The adjusted search terms as per searched electronic databases or search tools.**
(DOCX)

**S3 Table. Quality of the studies included in the systematic review and meta-analysis.**
(DOCX)

**S4 Table. Database.**
(XLSX)

**S5 Table. Meta-analysis database.**
(DOCX)

**S6 Table. R version 4.2.3. script.**
(DOCX)

**S1 Fig. Pooled prevalence of male sex in Peruvian patients with mpox.**
(TIF)

**S2 Fig. Pooled prevalence of female sex in Peruvian patients with mpox.**
(TIF)

**S3 Fig. Pooled prevalence of syphilis in Peruvian patients with mpox.**
(TIF)

**S4 Fig. Pooled prevalence of HIV in Peruvian patients with mpox.**
(TIF)

**S5 Fig. Pooled prevalence of Peruvian patients with HIV and mpox on highly active antiretroviral therapy (HAART).**
(TIF)

**S6 Fig. Pooled prevalence of Peruvian patients hospitalized for mpox.**
(TIF)

**S7 Fig. Pooled prevalence of heterosexual Peruvian patients with mpox.**
(TIF)

**S8 Fig. Pooled prevalence of Peruvian homosexual patients with mpox.**
(TIF)

**S9 Fig. Pooled prevalence of Peruvian bisexual patients with mpox.**
(TIF)

**S10 Fig. Pooled prevalence of fever in Peruvian patients with mpox.**
(TIF)

**S11 Fig. Pooled prevalence of headache in Peruvian patients with mpox.**
(TIF)

**S12 Fig. Pooled prevalence of myalgia in Peruvian patients with mpox.**
(TIF)

**S13 Fig. Pooled prevalence of fatigue in Peruvian patients with mpox.**
(TIF)

**S14 Fig. Pooled prevalence of local lymphadenopathy in Peruvian patients with mpox.**
(TIF)

**S15 Fig. Pooled prevalence of general lymphadenopathy in Peruvian patients with mpox.**
(TIF)

**S16 Fig. Pooled prevalence of lymphadenopathy of any type in Peruvian patients with mpox.**
(TIF)

**S17 Fig. Pooled prevalence of rash or skin lesions in consultation (local) in Peruvian patients with mpox.**
(TIF)

**S18 Fig. Pooled prevalence of rash or skin lesions in consultation (general) in Peruvian patients with mpox.**
(TIF)

**S19 Fig. Pooled prevalence of anogenital rash in Peruvian patients with mpox.**
(TIF)

**S20 Fig. Pooled prevalence of proctitis in Peruvian patients with mpox.**
(TIF)

## Author contributions

**Conceptualization:** Darwin A. León-Figueroa, Edwin Aguirre-Milachay, Milagros Diaz-Torres, Mario J. Valladares-Garrido.

**Data curation:** Darwin A. León-Figueroa, Edwin Aguirre-Milachay, Milagros Diaz-Torres, Virgilio E. Failoc-Rojas, Rodrigo Camacho-Neciosup, Mario J. Valladares-Garrido.

**Formal analysis:** Darwin A. León-Figueroa, Edwin Aguirre-Milachay, Milagros Diaz-Torres, Virgilio E. Failoc-Rojas, Rodrigo Camacho-Neciosup, Abel Eduardo Chávarry Isla, Mario J. Valladares-Garrido.

**Investigation:** Darwin A. León-Figueroa, Edwin Aguirre-Milachay, Milagros Diaz-Torres, Virgilio E. Failoc-Rojas, Rodrigo Camacho-Neciosup, Abel Eduardo Chávarry Isla, Mario J. Valladares-Garrido.

**Methodology:** Darwin A. León-Figueroa, Edwin Aguirre-Milachay, Milagros Diaz-Torres, Virgilio E. Failoc-Rojas, Rodrigo Camacho-Neciosup, Mario J. Valladares-Garrido.

**Project administration:** Darwin A. León-Figueroa, Edwin Aguirre-Milachay, Milagros Diaz-Torres, Virgilio E. Failoc-Rojas, Rodrigo Camacho-Neciosup, Abel Eduardo Chávarry Isla, Mario J. Valladares-Garrido.

**Resources:** Darwin A. León-Figueroa, Edwin Aguirre-Milachay, Milagros Diaz-Torres, Virgilio E. Failoc-Rojas, Rodrigo Camacho-Neciosup, Abel Eduardo Chávarry Isla, Mario J. Valladares-Garrido.

**Software:** Darwin A. León-Figueroa, Edwin Aguirre-Milachay, Milagros Diaz-Torres, Abel Eduardo Chávarry Isla, Mario J. Valladares-Garrido.

**Supervision:** Darwin A. León-Figueroa, Edwin Aguirre-Milachay, Milagros Diaz-Torres, Abel Eduardo Chávarry Isla, Mario J. Valladares-Garrido.

**Validation:** Darwin A. León-Figueroa, Edwin Aguirre-Milachay, Milagros Diaz-Torres, Virgilio E. Failoc-Rojas, Rodrigo Camacho-Neciosup, Abel Eduardo Chávarry Isla, Mario J. Valladares-Garrido.

**Visualization:** Darwin A. León-Figueroa, Edwin Aguirre-Milachay, Milagros Diaz-Torres, Virgilio E. Failoc-Rojas, Rodrigo Camacho-Neciosup, Abel Eduardo Chávarry Isla, Mario J. Valladares-Garrido.

**Writing – original draft:** Darwin A. León-Figueroa, Edwin Aguirre-Milachay, Milagros Diaz-Torres, Virgilio E. Failoc-Rojas, Rodrigo Camacho-Neciosup, Abel Eduardo Chávarry Isla, Mario J. Valladares-Garrido.

**Writing – review & editing:** Darwin A. León-Figueroa, Edwin Aguirre-Milachay, Milagros Diaz-Torres, Virgilio E. Failoc-Rojas, Rodrigo Camacho-Neciosup, Abel Eduardo Chávarry Isla, Mario J. Valladares-Garrido.

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
