## [Decision Letter · Decision Letter 0]

Dear Dr. Valladares-Garrido,

Thank you for submitting your manuscript to PLOS ONE. After careful consideration, we feel that it has merit but does not fully meet PLOS ONE’s publication criteria as it currently stands. Therefore, we invite you to submit a revised version of the manuscript that addresses the points raised during the review process.

We look forward to receiving your revised manuscript.

Kind regards,

Abdelaziz Abdelaal, M.D.

Academic Editor

PLOS ONE

**Journal Requirements:**

1. When submitting your revision, we need you to address these additional requirements. Please ensure that your manuscript meets PLOS ONE's style requirements, including those for file naming. The PLOS ONE style templates can be found at https://journals.plos.org/plosone/s/file?id=wjVg/PLOSOne_formatting_sample_main_body.pdf and https://journals.plos.org/plosone/s/file?id=ba62/PLOSOne_formatting_sample_title_authors_affiliations.pdf 2. We notice that your supplementary figures are uploaded with the file type 'Figure'. Please amend the file type to 'Supporting Information'. Please ensure that each Supporting Information file has a legend listed in the manuscript after the references list. 3. When completing the data availability statement of the submission form, you indicated that you will make your data available on acceptance. We strongly recommend all authors decide on a data sharing plan before acceptance, as the process can be lengthy and hold up publication timelines. Please note that, though access restrictions are acceptable now, your entire data will need to be made freely accessible if your manuscript is accepted for publication. This policy applies to all data except where public deposition would breach compliance with the protocol approved by your research ethics board. If you are unable to adhere to our open data policy, please kindly revise your statement to explain your reasoning and we will seek the editor's input on an exemption. Please be assured that, once you have provided your new statement, the assessment of your exemption will not hold up the peer review process. 4. As required by our policy on Data Availability, please ensure your manuscript or supplementary information includes the following:  A numbered table of all studies identified in the literature search, including those that were excluded from the analyses.   For every excluded study, the table should list the reason(s) for exclusion.   If any of the included studies are unpublished, include a link (URL) to the primary source or detailed information about how the content can be accessed.  A table of all data extracted from the primary research sources for the systematic review and/or meta-analysis. The table must include the following information for each study:  Name of data extractors and date of data extraction  Confirmation that the study was eligible to be included in the review.   All data extracted from each study for the reported systematic review and/or meta-analysis that would be needed to replicate your analyses.  If data or supporting information were obtained from another source (e.g. correspondence with the author of the original research article), please provide the source of data and dates on which the data/information were obtained by your research group.  If applicable for your analysis, a table showing the completed risk of bias and quality/certainty assessments for each study or outcome.  Please ensure this is provided for each domain or parameter assessed. For example, if you used the Cochrane risk-of-bias tool for randomized trials, provide answers to each of the signalling questions for each study. If you used GRADE to assess certainty of evidence, provide judgements about each of the quality of evidence factor. This should be provided for each outcome.   An explanation of how missing data were handled.  This information can be included in the main text, supplementary information, or relevant data repository. Please note that providing these underlying data is a requirement for publication in this journal, and if these data are not provided your manuscript might be rejected.   

**Additional Editor Comments:**

I commend the authors for their work. It was a good approach to stratify the data by the publication year.

However, there are some concerns raised by the reviewers. These should be properly addressed before further consideration.

In addition to these comments, please address the following:

1- Why are all the figures provided as supplementary S1-S2-S3, etc.?

2- There is a considerable amount of heterogeneity in most analyses. Have you considered doing sensitivity analyses? Galbraith plots? checking for outliers?

Reviewers' comments:

Reviewer's Responses to Questions

**Comments to the Author**

1. Is the manuscript technically sound, and do the data support the conclusions?

Reviewer #1: Yes

Reviewer #2: Yes

2. Has the statistical analysis been performed appropriately and rigorously?

Reviewer #1: Yes

Reviewer #2: Yes

3. Have the authors made all data underlying the findings in their manuscript fully available?

Reviewer #1: Yes

Reviewer #2: Yes

4. Is the manuscript presented in an intelligible fashion and written in standard English?

Reviewer #1: Yes

Reviewer #2: No

**Reviewer #1: ** REVIEW

Overall assessment;

This manuscript has attempted to provide a comprehensive analysis of the epidemiological and clinical characteristics of mpox in Peruvian patients. While the study contributes valuable data to the growing body of literature on mpox in Latin America, there are several critical issues regarding methodological rigor, clarity of discussion, and the interpretation of findings. Below is a detailed evaluation of the manuscript’s strengths and weaknesses.

Major concerns:

1. Methodological limitation

• The retrospective nature of most included studies is acknowledged but not critically analyzed in areas such as recall bias, missing data, and confounding variables that could have influenced the reported epidemiological trends.

2. Overgeneralization and lack of nuanced interpretation

• The authors heavily emphasize that the predominance of mpox cases in men, particularly MSM, is consistent with global trends. However, the discussion lacks a critical analysis of potential underreporting or diagnostic biases among women and other populations. Are women less affected, or is their presentation underrecognized and underreported?

•The claim that transgender women are at higher risk than cisgender women lack adequate contextualization. The authors cite studies suggesting differential risks but do not explore how societal, behavioral, and healthcare access factors contribute to these disparities.

•The assertion that mpox severity is strongly correlated with HIV status is valid but lacks differentiation between well-controlled HIV infection and advanced HIV/AIDS. Instead of using ART access as a measure of how controlled HIV infection is (which may be flawed if viral load is higher), the discussion can be more robust by explicitly distinguishing between HIV+ individuals with undetectable viral loads (or a categorical spectrum thereof) and those with significant immunosuppression (low CD4/high viral load).

3. Inconsistencies in clinical data reporting:

•The discussion on hospitalization rates compares the Peruvian cohort (5%) with a Brazilian study (10.5%) without acknowledging potential confounders such as differences in healthcare infrastructure, diagnostic criteria, and thresholds for hospitalization.

4. Citations issues:

•The manuscript cites multiple systematic reviews and meta-analyses (e.g., Ortiz-Saavedra et al., Benites-Zapata et al…etc), but the authors do not critically engage with these studies’ limitations, such as sample heterogeneity or publication bias.

Minor concerns:

1. Lack of clarity in writing and organization:

•Some sections contain redundant information. For example, the discussion of HIV comorbidity appears in multiple parts of the text, leading to somewhat unnecessary overemphasized repetition.

•The manuscript would benefit from clearer subheadings within the discussion section to separate epidemiological trends, clinical presentation, and disease severity.

2. Limited discussion on public health implications:

•While the study discusses clinical manifestations, it does not sufficiently explore public health interventions. How should these findings inform mpox prevention strategies in Peru? What are the implications for vaccination, contact tracing, and community engagement?

•The manuscript briefly mentions that "intervention studies focused on prevention and treatment in HIV co-infected patients could also offer key strategies to mitigate the effects of future outbreaks." This is a crucial point that deserves further expansion.

•The conclusions should be refined to offer more actionable public health recommendations. The conclusion that these findings support prevention strategies is overly broad and lacks specificity. What exact strategies should be prioritized based on this data?

Future directions

•The call for longitudinal studies is appropriate, but the authors should specify which knowledge gaps remain unaddressed.

Other comments.

1.Confidence intervals are still wide for some estimates (thanks for sensitivity analysis), yet their implications are not discussed. For example, the estimate for lymphadenopathy is 83% (95% CI: 25-100%), making these pooled results highly questionable.

2.S-table 3 talks about the quality of the dengue studies included in the review. This seems unusual as there is no relevance to dengue for this paper. An explanation could help the reader to understand.

3.The choice of databases includes some that are not standard for systematic reviews (e.g., Google Scholar, which lacks robust indexing and reproducibility). The inclusion of these databases should be justified.

My recommendation is that this manuscript provides valuable epidemiological insights into mpox in Peru. However, substantial revisions are necessary to enhance methodological transparency, refine the discussion of key findings, and strengthen the manuscript’s public health implications. Addressing these concerns will significantly improve its scientific rigor and impact. Specifically, by address the raised issues above including refining the discussion, ensuring more rigorous citation practices, improving clarity and structure, and expanding the public health relevance will enhance the manuscript’s contribution to the infectious disease research community.

**Reviewer #2: ** The article shows important descriptive data that can be useful to understand the characteristics of mpox in the context of Peru and Latin American. However, I consider that some adjustments are necessary prior to publication.

1) Abstract

"The main objective of this study is to determine the prevalence of epidemiological and clinical characteristics of Peruvian patients diagnosed with mpox, providing a detailed view of the situation of this affected population."

- I suggest changing this phrase to "The main objective of this study is to determine the epidemiological and clinical characteristics of Peruvian patients diagnosed with mpox, providing a detailed view of the situation of this affected population."

2) Introduction

The data of Table 1 don't add much to the Introduction where they are currently placed. The characteristics described are not mentioned in the Introduction and neither in the Discussion. It would be interesting to compare them with the results of the author's review in the Discussion.

3) Methods

- Explain the reason for excluding randomized clinical trials of the review.

4) Results

- The titles of Tables 2 and 3 should inform that they are presenting data from the observational studies. This is important because the data of the case series and case reports are presented separately in Table 4. The titles of the tables should allow understanding independently, without the necessity of referring to the main text.

"On the other hand, the fact that 91% of coinfected patients were receiving antirretroviral therapy suggests that, although access to treatment is adequate, these individuals remain susceptible to mpox infection, raising questions about the impact of relative immunosuppression on the clinical course of the disease."

- I suggest rephrasing this paragraph. The presence of HIV coinfection does not change the susceptibility to mpox infection, since several immunocompetent individuals were also infected. If severe immunosuppression is present, it can alter the severity of mpox, but the study does not address outcomes. Hospitalization is discussed in later paragraphs, what I consider important to maintain. Possible reasons for the finding of the high prevalence of HIV coinfection among the mpox cases should be discussed.

- Data from the observational studies were analyzed separately from the ones of the case series and reports. I understand from reading the manuscript that individual data were not available for the four observational studies included in the review. If this assumption is correct, this should be stated in the text or, if it's not, the reason why pooled individual data analysis was not performed in variables such as sex or HIV status.

**Do you want your identity to be public for this peer review?** For information about this choice, including consent withdrawal, please see our Privacy Policy

Reviewer #1: No

Reviewer #2: No

---

## [Author Response · Author response to Decision Letter 1]

7 Apr 2025

Dear Editor,

Thank you very much for reviewing our article, "Epidemiological and clinical characteristics of Peruvian patients with mpox: a systematic review and meta-analysis". Your suggestions and comments will be addressed below. Thank you for your valuable time and excellent review.

Editor's comments

1. 1. When submitting your revision, we need you to address these additional requirements. Please ensure that your manuscript meets PLOS ONE's style requirements, including those for file naming. The PLOS ONE style templates can be found at https://journals.plos.org/plosone/s/file?id=wjVg/PLOSOne_formatting_sample_main_body.pdf and https://journals.plos.org/plosone/s/file?id=ba62/PLOSOne_formatting_sample_title_authors_affiliations.pdf

Our response: Thank you for your feedback. We have reviewed and adjusted our manuscript in accordance with the guidelines provided by PLOS ONE. Additionally, we have ensured that all files meet the required formatting standards, including the proper file naming.

Attached is the revised version of the manuscript, now aligned with the specified guidelines, including the correct format for the title, authors, affiliations, and the main body of the article.

2. 2. We notice that your supplementary figures are uploaded with the file type 'Figure'. Please amend the file type to 'Supporting Information'. Please ensure that each Supporting Information file has a legend listed in the manuscript after the references list.

Our response: Thank you for your comments. We have updated the file type to "Supplementary Information," as requested.

Additionally, we have ensured that each supplementary file is referenced in the manuscript. The captions for each figure are not specified separately, as they are included in full within the main text of the article.

Our response: We have modified the data availability statement. All our data are included in the article and its supplementary materials.

4. 4. As required by our policy on Data Availability, please ensure your manuscript or supplementary information includes the following:

Our response: The availability of the data was verified, and it is detailed in the article and supplementary material.

Our response: The numbered table of all the studies identified in the bibliographic search has been presented. Please verify S4 Table.

Our response: The excluded studies have been presented along with the reasons for exclusion. Please verify S4 Table.

Our response: The included studies were verified, and all are published.

Our response: Your recommendation is included in the article and supplementary material. Please review Tables 2, 3, 4, and S5 Table.

Our response: The initials of the data extractors and the date of data extraction were added.

Our response: Eligibility criteria were established to confirm that the study met the requirements to be included in the review.

Our response: All extracted data, as well as the included and excluded articles, are reported in detail in the study and supplementary material. It is recommended to review the documents and tables provided. Our study can be replicated, as it presents the R codes used in detail, along with tables containing the data entered into the analysis.

Our response: All supporting information and data are detailed in the supplementary material and do not require permission, as they were created by the authors based on published articles.

Our response: Our study and its analysis include a table that presents the complete risk of bias and quality/uncertainty assessments for each study or outcome using the JBI tool. Additionally, publication bias was assessed using a funnel plot and the Egger test. Review supplementary material.

Our response: All the data extracted from the articles are presented in detail in the main article and supplementary material.

Our response: We agree with your comment. All the data have been provided in the article and supplementary materials.

16. Additional Editor Comments: I commend the authors for their work. It was a good approach to stratify the data by the publication year. However, there are some concerns raised by the reviewers. These should be properly addressed before further consideration.

Our response: Thank you for your valuable comments. We have addressed all the reviewers' observations and suggestions in detail, ensuring that the necessary improvements have been incorporated into the document.

17. In addition to these comments, please address the following: 1- Why are all the figures provided as supplementary S1-S2-S3, etc.?

Our response: The figures in the supplementary material (S1, S2, S3, etc.) provided serve as evidence for the synthesis in Table 5. These figures support the prevalence of the epidemiological and clinical characteristics of Peruvian patients with mpox.

The author guidelines specify a set number of figures, which is why they were included in the supplementary material.

18. 2- There is a considerable amount of heterogeneity in most analyses. Have you considered doing sensitivity analyses? Galbraith plots? checking for outliers?

Our response: We appreciate your comment regarding the considerable heterogeneity observed in the analyses. We agree with your observation and have addressed this heterogeneity in the limitations of the study, providing an explanation for it. Although we considered conducting a sensitivity analysis, it was not feasible due to the variability in the available data.

Nevertheless, we established strict criteria for our systematic review, following PRISMA guidelines. Additionally, we developed a detailed protocol, devised a rigorous search strategy, and employed methods to assess the quality of evidence and publication bias.

Dear Editor, all reviewer comments have been addressed. The English grammar has been improved, and the article has been formatted according to the specific guidelines of Plos One. Additionally, the epidemiological data has been updated in line with the latest information from the World Health Organization (WHO).

Reviewer #1:

Overall assessment;

This manuscript has attempted to provide a comprehensive analysis of the epidemiological and clinical characteristics of mpox in Peruvian patients. While the study contributes valuable data to the growing body of literature on mpox in Latin America, there are several critical issues regarding methodological rigor, clarity of discussion, and the interpretation of findings. Below is a detailed evaluation of the manuscript’s strengths and weaknesses.

Our response: We greatly appreciate your review of our article, dear doctor. We have addressed all the observations and accepted the recommendations provided.

1. Reviewer says: 1. Methodological limitation

• The retrospective nature of most included studies is acknowledged but not critically analyzed in areas such as recall bias, missing data, and confounding variables that could have influenced the reported epidemiological trends.

Our response: We deeply appreciate the reviewer’s observation and fully agree with it. We acknowledge that most of the included studies have a retrospective design, which introduces several factors that could have influenced the observed results and epidemiological trends. In fact, we assessed the quality of the included studies, which allowed us to identify potential limitations inherent to these designs.

In the study limitations section, we have highlighted these critical aspects, particularly recall bias, missing data, and confounding variables. Recall bias may have affected the accuracy of the data, as participants or retrospective records could have had difficulties accurately remembering past events. Additionally, missing data (e.g., loss to follow-up or incomplete information) may have introduced further biases that impacted the validity of the results. Finally, confounding variables, such as uncontrolled or inadequately identified factors in the studies, could have influenced the observed trends and distorted the conclusions.

2. Reviewer says: 2. Overgeneralization and lack of nuanced interpretation

• The authors heavily emphasize that the predominance of mpox cases in men, particularly MSM, is consistent with global trends. However, the discussion lacks a critical analysis of potential underreporting or diagnostic biases among women and other populations. Are women less affected, or is their presentation underrecognized and underreported?

Our response: Thank you for your valuable observation. In response, we have added a paragraph in the Discussion section that addresses the potential underreporting and diagnostic biases affecting women and other populations. The revised text acknowledges that while current data indicate a predominance of mpox cases among men, particularly MSM, this pattern may not reflect the actual burden in other groups. We now emphasize the importance of inclusive surveillance strategies and a more nuanced interpretation of the data to better capture the epidemiology of mpox across diverse populations.

3. Reviewer says: •The claim that transgender women are at higher risk than cisgender women lack adequate contextualization. The authors cite studies suggesting differential risks but do not explore how societal, behavioral, and healthcare access factors contribute to these disparities.

Our response: Thank you for your comments. We have reviewed and modified the manuscript to more thoroughly address the societal, behavioral, and healthcare access factors contributing to the disparities in infection risks between transgender women and cisgender women. Following your recommendations, we expanded the discussion based on the cited articles, explaining how social stigma, discrimination, the higher prevalence of HIV in transgender women, and barriers to healthcare access contribute to their increased vulnerability to mpox infection. We believe these changes provide better contextualization of the elevated risk faced by transgender women and offer a more comprehensive analysis of the factors influencing these disparities.

4. Reviewer says: •The assertion that mpox severity is strongly correlated with HIV status is valid but lacks differentiation between well-controlled HIV infection and advanced HIV/AIDS. Instead of using ART access as a measure of how controlled HIV infection is (which may be flawed if viral load is higher), the discussion can be more robust by explicitly distinguishing between HIV+ individuals with undetectable viral loads (or a categorical spectrum thereof) and those with significant immunosuppression (low CD4/high viral load).

Our response: Thank you for your valuable comments. In response to your observation, we have reorganized and expanded the discussion following your recommendations. An explicit distinction has been made between individuals with well-controlled HIV, characterized by undetectable viral loads and adequate CD4 counts, and those with advanced HIV or AIDS, who exhibit significant immunosuppression, marked by low CD4 counts and high viral loads. This revision makes the discussion on the correlation between mpox severity and HIV status more accurate and robust, addressing the impact of immunosuppression on the clinical course of the disease.

5. Reviewer says: 3. Inconsistencies in clinical data reporting:

•The discussion on hospitalization rates compares the Peruvian cohort (5%) with a Brazilian study (10.5%) without acknowledging potential confounders such as differences in healthcare infrastructure, diagnostic criteria, and thresholds for hospitalization.

Our response: Thank you for your valuable feedback. In response to your observation, we have revised the discussion to address potential confounders that may explain the differences in hospitalization rates between the Peruvian and Brazilian cohorts. We acknowledge that factors such as differences in healthcare infrastructure, diagnostic criteria, and thresholds for hospitalization could influence these rates.

6. Reviewer says: 4. Citations issues:

•The manuscript cites multiple systematic reviews and meta-analyses (e.g., Ortiz-Saavedra et al., Benites-Zapata et al…etc), but the authors do not critically engage with these studies’ limitations, such as sample heterogeneity or publication bias.

Our response: Thank you for your valuable comments. We have added and highlighted the recommended changes, acknowledging that it is crucial to consider the limitations of the cited studies, such as sample heterogeneity due to variations in populations, regions, and clinical settings, which can affect the generalizability of the results. Additionally, we addressed the potential publication bias, which could overrepresent studies with positive results, influencing the overall conclusions. These considerations have been incorporated into the discussion to provide a more nuanced and accurate view of the findings.

7. Reviewer says: 1. Lack of clarity in writing and organization:

•Some sections contain redundant information. For example, the discussion of HIV comorbidity appears in multiple parts of the text, leading to somewhat unnecessary overemphasized repetition.

Our response: Thank you for your comments. In response to your observation, we have reviewed and revised the manuscript to improve clarity and organization, eliminating redundancies in the discussion of HIV comorbidity. The information has been restructured to avoid unn

---

## [Decision Letter · Decision Letter 1]

Epidemiological and clinical characteristics of Peruvian patients with mpox: a systematic review and meta-analysis.

PONE-D-24-53481R1

Dear Dr. Valladares-Garrido,

We’re pleased to inform you that your manuscript has been judged scientifically suitable for publication and will be formally accepted for publication once it meets all outstanding technical requirements.

Kind regards,

Miquel Vall-llosera Camps

Staff Editor

PLOS One

Reviewers' comments:

Reviewer's Responses to Questions

**Comments to the Author**

Reviewer #1: All comments have been addressed

Reviewer #2: All comments have been addressed

2. Is the manuscript technically sound, and do the data support the conclusions?

Reviewer #1: (No Response)

Reviewer #2: Yes

3. Has the statistical analysis been performed appropriately and rigorously?

Reviewer #1: (No Response)

Reviewer #2: Yes

4. Have the authors made all data underlying the findings in their manuscript fully available?

Reviewer #1: (No Response)

Reviewer #2: Yes

5. Is the manuscript presented in an intelligible fashion and written in standard English?

Reviewer #1: (No Response)

Reviewer #2: Yes

Reviewer #1: (No Response)

Reviewer #2: (No Response)

**Do you want your identity to be public for this peer review?** For information about this choice, including consent withdrawal, please see our Privacy Policy

Reviewer #1: No

Reviewer #2: No

---

## [Editor Report · Acceptance letter]

PONE-D-24-53481R1

PLOS ONE

Dear Dr. Valladares-Garrido,

I'm pleased to inform you that your manuscript has been deemed suitable for publication in PLOS ONE. Congratulations! Your manuscript is now being handed over to our production team.

Kind regards,

on behalf of

Dr. Miquel Vall-llosera Camps

Staff Editor

PLOS ONE